# Current Stage of Marine Ceramic Grafts for 3D Bone Tissue Regeneration

**DOI:** 10.3390/md17080471

**Published:** 2019-08-15

**Authors:** Patricia Diaz-Rodriguez, Miriam López-Álvarez, Julia Serra, Pío González, Mariana Landín

**Affiliations:** 1R + D Pharma Group (GI-1645), Department of Pharmacology, Pharmacy and Pharmaceutical Technology, School of Pharmacy, Universidade de Santiago de Compostela, 15782 Santiago de Compostela, Spain; 2Department of Chemical Engineering and Pharmaceutical Technology, School of Sciences, Universidad de La Laguna (ULL), Campus de Anchieta, 38200 La Laguna (Tenerife), Spain; 3New Materials Group, Department of Applied Physics, University of Vigo, IISGS, MTI-Campus Lagoas-Marcosende, Vigo 36310, Spain

**Keywords:** marine ceramic grafts, calcium phosphate, bone tissue regeneration, osteoinductive properties, structural strength, dissolution rate

## Abstract

Bioceramic scaffolds are crucial in tissue engineering for bone regeneration. They usually provide hierarchical porosity, bioactivity, and mechanical support supplying osteoconductive properties and allowing for 3D cell culture. In the case of age-related diseases such as osteoarthritis and osteoporosis, or other bone alterations as alveolar bone resorption or spinal fractures, functional tissue recovery usually requires the use of grafts. These bone grafts or bone void fillers are usually based on porous calcium phosphate grains which, once disposed into the bone defect, act as scaffolds by incorporating, to their own porosity, the intergranular one. Despite their routine use in traumatology and dental applications, specific graft requirements such as osteoinductivity or balanced dissolution rate are still not completely fulfilled. Marine origin bioceramics research opens the possibility to find new sources of bone grafts given the wide diversity of marine materials still largely unexplored. The interest in this field has also been urged by the limitations of synthetic or mammalian-derived grafts already in use and broadly investigated. The present review covers the current stage of major marine origin bioceramic grafts for bone tissue regeneration and their promising properties. Both products already available on the market and those in preclinical phases are included. To understand their clear contribution to the field, the main clinical requirements and the current available biological-derived ceramic grafts with their advantages and limitations have been collected.

## 1. Introduction

Cell culture in two dimensions has traditionally been used to test the biological response to different biomaterials or to perform drug screening by growing cells on flat surfaces. These conventional cell monolayer cultures do not fully reflect the essential physiology of real tissues as they modify the tissue-specific architecture (forced polarity, flattened cell shape), mechanical/biochemical signals, and cell-to-cell communications [1,2]. Therefore, the simplicity of this method supposes an advantage but, at the same time, a disadvantage since the “in vivo” 3D complex environment is not represented, resulting in non-predictive data for clinical applications [3]. 

Current alternatives include three-dimensional cell culture techniques based on organoids, cell encapsulation on hydrogels, or growing the cells on scaffolds instead of flat surfaces. Organoids are 3D multicellular tissue constructs that can growth either in gels or in suspension generating organ-like structures [4]. Cell encapsulation allows cell survival and extracellular matrix deposition while permitting the analysis of complex cell interactions [5]. On the other hand, scaffolds provide convenient cell support due to their porosity, facilitating oxygen, nutrients, and waste transportation [6]. These systems resemble more closely the tissue physiological characteristics by providing structural tensile strength, cell adhesion, polarity, migration, and proliferation [7]. 

Moreover, for tissue regeneration purposes, scaffolds can be implanted to help tissue reconstruction, and then be either removed or biodegraded after fulfilling its purpose [8]. Bone graft materials or void fillers are used as direct cell support in certain patients to promote regeneration [9]. They are usually based on ceramics as calcium phosphate (CaP) grains of different sizes (in dentistry around 0.5–1 mm) depending on the bone defect volume to be filled. When bone grafts are placed into the defect a 3D scaffold with intergranular porosity is generated in addition to the intrinsic porosity of the grains. These bioceramics are commonly tested in the laboratory in 3D cell culture by their seeding using variable defect volumes to obtain cellular responses closer to the clinic [8,10].

An intense research interest exists on bioceramic structure, including combinations of different CaP phases (hydroxyapatite (HA), beta-tricalcium phosphate (β-TCP)) to provide the optimal balance in stability/resorbability able to ensure bone regeneration. This need promotes searching for new bioceramic sources including marine derived structures with their chemical and morphological particularities. These bioceramics should follow the ethics requirements, the animal welfare awareness (avoiding animal sacrifices), as well as sustainability regulations. Therefore, the use of animal discards and fishing by-products or wastes as calcium phosphate sources has recently gained increased attention. Finally, 3D composite materials by including ceramic granules into polymeric hydrogels, typically collagen, with or without growth factors are also used. Traditional chemical methodologies or advanced ones, such as 3D printing, are being investigated [10].

The present review covers the current stage of the main marine origin bioceramic grafts tested in 3D in vitro culture and designed for bone tissue regeneration including their promising properties. Both products already available on the market and those in preclinical phases are included. To understand their clear contribution to the field, main clinical requirements and current available biological-derived ceramic grafts, with their advantages and limitations, have been collected. 

## 2. Bone Grafts: Limitations of Auto and Allografts

Bone-related diseases suppose nowadays one of the main causes of disability and involve a high number of surgical interventions. Many of the required interventions are caused by pathologies related to aging, associated with the large number of cellular changes [11]. Osteoarthritis and osteoporosis are the age-related diseases with the highest incidence in the field of traumatology. Moreover, bone defects can also be derived from car accidents, falls or sport-related injuries, trauma or tumors, and congenital diseases, such as spinal fractures, deterioration of intervertebral discs, and narrowing of spinal canal (stenosis). The dentistry field must also be considered where, again aging, genetic factors, incorrect oral hygiene, or oral trauma, influence the appearance of periodontitis and caries which, at the same time, will contribute to the increase of partial or total edentulism incidence, including alveolar bone resorption. All these bone alterations require the use of grafts to promote functional tissue recovery [12]. This high demand of bone tissue implies over two million of bone grafting procedures performed annually worldwide [13] with an estimated global market of $2.6B [14]. 

The ideal bone substitute, according to the clinicians, should be biocompatible, structurally similar to bone, easily moldable within the osseous defect, not produce inflammatory response, osteoconductive, osteoinductive, and resorbable. It must also be non-thermally conductive, sterilizable, and easily accessible at a reasonable cost [15,16,17,18]. In terms of porosity, the recommended pore size for the growth of capillaries is 50 μm, and a size of 200 µm is needed for the growth of new osteons in the pores [19,20]. It is important to consider the colonization of these macropores by mesenchymal cells, allowing bone apposition, begins at two or three weeks post-implantation, so the resorption of the graft should not be too fast [20]. Moreover, micropores of less than 10 μm allow circulation of organic liquids and diffusion of substances through the matrix while increasing the specific surface area, improving the metabolic environment for bone-producing cells, and accelerating remodeling. Finally, the presence of interconnected porosity favors the appearance of a capillary force that actively absorbs the patient’s blood and bone marrow into the matrix of the material. Furthermore, to be able to talk about osteoconduction, porosity must allow vascularization and cell growth, also having a crucial role in graft biodegradation [21,22]. 

Autografts remain as the gold standard in bone regeneration for both trauma and dentistry given their osteoinductive properties. In the case of traumatology, grafts are usually obtained from non-essential bone such as the iliac crest or the fibula [15]. In dentistry, they are obtained from intraoral sites for small defects (chin, maxillary tuberosity, ascending branch) or from extraoral sites when a higher volume is required (iliac crest, tibia, or calotte). The choice will depend on the type, size, and shape of the bone cavity, clinical experience, and professional preference. Autogenous cancellous bone has greater osteogenic capacity while cortical bone provides greater stability [23]. Autografts have disadvantages such as postoperative morbidity of the donor area with bruising, residual pain, fractures, infection, hemorrhage, muscle weakness, neurological injury, and scarring [24,25]. Besides, their availability is limited as, in some cases, the amount of graft extracted is insufficient. On the other hand, the considerable increase in surgical time and the requirement of an additional anesthetic procedure also imply clear limitations [25].

On the other hand, allografts result from bone tissue obtained from another human, which can be alive or a cadaveric donor (bone donation), and they must be processed in a tissue bank [15]. Allografts can be used mineralized (tissue bank) or demineralized (after being subjected to different chemical processes). These latter are the ones usually commercialized under various trademarks (see Table 1). Mineralized allografts, preserved under freezing, show the advantages of autografts but with a slightly lower osteoinductive capacity due to the loss of growth factors and inferior mechanical properties. Allografts present high availability, important quantities, and different shapes and sizes, avoiding sacrifices of host structures and donor site morbidity together with a reduced surgical time [13]. They are widely used in osteoarticular reconstructive surgery, in hospitals with a bone bank, given their ease of use and good results in bone fractures and defects filling [26,27]. They are also the most common type currently used in foot and ankle surgery [28]. Their limitations are related to the quality of the regenerated bone tissue that is not always predictable due to its limited osteoinduction, in addition to requiring costly and laborious processing to eliminate its antigenic capacity [23]. They also present limitations in their mechanical resistance and the potential risk of disease transmission from the donor such as HIV, hepatitis B, hepatitis C, and human T-lymphotropic virus [29]. 

Demineralized allografts or demineralized bone matrices (DBMs) are composed of the organic matrix of human bone: mainly collagen and inherent growth factors including bone morphogenic proteins (BMPs) stored in the tissue after the removal of at least 40% of the mineral content [13]. DBMs maintain the collagen matrix, which reproduces the three-dimensional architecture of bone facilitating and guiding the invasion, growth, and differentiation of the host cells [13]. They are thought to present superior osteoinductive capacity to that of mineralized allografts due to the greater availability of bone morphogenetic protein and growth factors. These non-bioceramic grafts maintain part of the structure and components of the mineralized tissue of origin and, in certain products, a residual mineral content. They are, however, limited to be used in bone defects without load bearing given their low biomechanical resistance [30]. Moreover, their osteoinductive capacity varies depending on donor age, mineral content, nature, sterilization and processing, receptor species (more osteoinduction in animal models than in humans), recipient region, or implant site. Removal or inactivation of viruses must be performed to avoid disease transmission [31].

Demineralized allografts are used combined with other compounds such as autologous bone, allografts, blood and autologous marrow or with synthetic materials. Numerous studies suggest that DBM enriched with bone marrow could be comparable to autografts to treat long bone fractures [32,33,34]. Other possibilities include, outside the scope of this review, the use of bone morphogenetic proteins in collagen sponges, these molecules are considered the most potent inducers of bone consolidation. An example of a commercial product is Infuse^TM^ Bone Graft (Medtronic), which consists on the recombinant human BMP-2 applied to an absorbable collagen sponge carrier (bovine origin). It is indicated for non-load applications on spinal fusion procedures in skeletally mature patients, open tibia shaft fractures, and oral-maxillofacial procedures as an alternative to autologous bone grafts. Contraindications include patients with hypersensitivity to any of the components, skeletally immature, with active infections at the surgical site, with inadequate neurovascular status or in the vicinity of resected or existing tumors, among others [35].

Biological grafts of human origin, reviewed in this section, meet the osteoconduction requirement, especially autografts and mineralized allografts. In the case of the autografts, limited to small defects, they also provide certain osteoinduction and osteogenesis. In order to overcome the limited availability of autografts, and the rest of the already mentioned disadvantages, when larger volumes are required for applications requiring loads, CaP grafts of mammalian origins are also available on the market. 

## 3. Bioceramic Xenografts: Mammalian Origin

The bone tissue physiological similarities between humans and mammals make easy to consider the mammalian mineral tissue as an adequate source of bone xenografts for human use. To develop these xenografts, donors’ bone is treated using different protocols to avoid disease transmission and immunologic reactions while ensuring biocompatibility and restoring bone structure and function [36]. A good number of xenografts are obtained after bone tissue treatment with the BioCleanse® process. This process consists of the tissue sterilization and cleansing procedure using low temperature by combining mechanical and chemical processes removing cells, lipids, and other sources of antigenic material [37,38]. However, other companies decide to use their proprietary manufacturing process. Xenografts can not only be used alone but also combined with autografts, as mentioned above, to decrease the amount of autogenous bone needed reducing patient morbidity and showing improved regeneration when compared to xenografts alone [39,40].

Bovine based grafts are the most commonly used bone xenografts in orthopedic surgery [41]. Several bovine based bone grafts are already on the market with variable preparations, from blocks to granules, shown in Table 2. Moreover, the close genotype between humans and pigs have made this mammal donor another commonly used source for bone grafts with similar results to bovine xenografts [42,43,44]. The nowadays commercialized porcine bone grafts are also described in Table 2. Equine bone tissue is also used as source of bone grafts. These mammal derived grafts are thought to have osteogenic and bone inductive properties to assist bone healing and can be incorporated into the bone host tissue acting as support for bone colonization. However, they do not allow remodeling and they stay mainly unaltered on the host bone [45]. To allow osteoclast function, the addition of collagen could promote graft resorption [45]. In agreement with this, the use of collagenated porcine bone grafts indicated the possibility of graft resorption [45]. Moreover, the coating of bovine graft with poly(l-lactide-co-ɛ-caprolactone) (PLCL) and polysaccharides promoted an increased proliferation of mesenchymal stem cells and bone formation when compared to un-treated bovine bone grafts [46]. Furthermore, the use of collagenated porcine bone xenografts showed better clinical results in ridge preservation procedures when compared to cortical porcine bone [47]. Already commercialized combinations of ceramic bone xenografts with polymers or extracellular matrix proteins are shown Table 2 in the last four rows. In general, bone xenografts present a high success rate without major complications after use [48].

### Mammalian Xenografts in Research

In addition to the already commercialized bone grafts from mammal origin, new approaches have been proposed to enhance their osteoconductivity. To this end, several strategies have been explored. Park and coworkers modified the surface of deproteinized porcine cancellous bone to introduce magnesium ions on their surface. The resulting bone xenograft showed an apparent increase in bone ingrowth when compared to non-treated deproteinized bovine and porcine grafts [49]. In a similar way, fluoride ions were added to porcine bone xenografts leading to an increase in mesenchymal stem cell proliferation and osteogenic differentiation together with an accelerated “in vivo” bone ingrowth [50]. Additionally, different authors added polymers and proteins as collagen to bioceramic xenografts to increase their hydrophilicity and improve mechanical properties [51]. High molecular weight hyaluronic acid combined with bovine xenografts has shown an increase in bone healing when compared to bovine xenografts alone [52]. Moreover, the addition of biomembrane fraction 1 protein, obtained from latex, to bone grafts was able to modulate the expression of extracellular matrix degradative enzymes “in vivo” [53]. Another attempt to enhance bone regeneration has been the implantation of bovine xenografts after soaking with “*Hypericum perforatum”* extract leading to an improvement in bone healing [54].

Other research aims in the mammalian xenograft field are to obtain and characterize other tissue donors suitable as bone xenografts for humans. One example is the use of calcinated antler cancellous bone obtained from deer (*Cervidae* spp.) due to their similar structure and composition when compared to human bone [55]. Moreover, antlers’ growth cycle guarantees a high annual availability of the graft source since the animal discards them each year and, therefore, their use does not involve animal sacrifice promoting the re-use of waste and discards [56,57,58]. Zhang and coworkers [59] demonstrated the utility of these grafts for bone regeneration, inducing neovascularization and osteogenesis in mandible defects at similar levels to grafts already on the market. As for all the animal donors, there are several differences between donor characteristics such as age that modulate xenografts porosity and crystallinity [60]. 

## 4. Bioceramic Xenografts: Marine Origin

The ocean provides bioceramics with interconnected porosity in a hierarchical structure similar to those of trabecular human bone, making them suitable materials as bone grafts with osteoconductive properties [61]. One example is the exoskeleton of several coral species, mainly composed of the crystalline ceramic structure aragonite (calcium carbonate). Two reef-building coral skeletons are commercially used as bone grafts, Porites and Goniopora, given their availability in large quantities and highly consistent structure. These bioceramics are subjected to thermal treatments to avoid immunogenic responses with only tiny quantities of intra-crystalline proteins remaining. However, they possess inherent weakness in compression and the absorption of calcium carbonate is too quick, limiting the use of these grafts [61,62]. 

To increase the strength of coral skeletons, so they can support the high compressive forces exerted in load bearing long bones, a chemical transformation can be performed from the native calcium carbonate composition to hydroxyapatite by a hydrothermal conversion [63]. This procedure increases graft durability as the resulting hydroxyapatite degrades slowly, with complete resorption achieved after a year or even longer. In fact, a clinical evaluation by Korovessis and coworkers in dorsal and lateral fusion for degenerative lumbar spine disease, proved the complete resorption of this coralline hydroxyapatite (mixed with local bone and bone marrow) one year after surgery [64]. This biomaterial was also tested by Coughlin and coworkers in hindfoot arthrodesis concluding their effectiveness as a bone graft in clinical foot procedures [65]. However, they reported difficulty of containing it, with extrusion present in all patients, and a too slow resorption rate (graft presence up to 6 years after surgery). More recently, Messina provided a recipient site surgical preparation protocol in dental implantation using coralline HA granules and homologous fibrin glue to minimize risks of failure related to mechanical instability and low retention at the surgical site [66]. 

Another variant already on the market consists of grafts produced from converted coral skeletons not entirely transformed to hydroxyapatite, so that some parts remain as calcium carbonate. In this way, the biodegradation properties are improved to suit bone remodeling, turnover, and natural bone healing [63]. Finally, to overcome the contaminant issues (intracrystalline proteins that accompany biogenic crystals) and limit ecological impact, farming techniques are being implemented, growing these marine structures in artificial aquaria [62]. The coral growth in aquariums will allow the strict control of conditions and the addition of determined ions of interest during the growth period (as silicate or phosphate ions). This technique allows to enrich the calcium carbonate structure obtaining higher bioactivity at the final mineral product [67]. The commercial coralline-derived grafts currently available for clinicians are shown at Table 3.

There are products on the market that combine DBM allografts with coralline-graft granules, as is the case of StaGraft^TM^ DBM PLUS (Zimmer Biomet) with coralline hydroxyapatite/calcium carbonate granules of Pro-Osteon® 500R. These products are indicated for filling bony voids or gaps in extremities and pelvis that are not intrinsic to the bony stability of the structure, autograft extender in the spine, bone void filler in the spine (posterolateral spine), and craniofacial defects fillers in craniotomies no larger than 25 cm^2^.

### Marine Xenografts in Research

The commercially available unconverted coral (calcium carbonate) is still not ideal for most long-term implant purposes due to its very fast dissolution rate and poor longevity and stability. Moreover, coralline grafts with partial conversion of coral to hydroxyapatite provide poor mechanical properties for load-bearing applications when high structural strength is required. Therefore, new developments have been investigated, such as a complete conversion of coral to pure hydroxyapatite and its subsequent coating with a sol-gel-derived HA to cover micro- and nano-pores within the intra-pore material, whilst maintaining the large pores [71]. Following this strategy, the biaxial strength was improved two-fold providing enhanced durability, longevity, and strength in the physiological environment for load-bearing bone graft applications where high strength requirements are required. In this case, the coralline origin is practically anecdotal as, apart from compositional modifications, morphological changes on the porosity of the original structure are also performed. Other previous attempts focused on the mechanical strength improvement were based on the preparation of fluorine- and zirconia-doped coralline HA [72], and the recent incorporation of Sr ions on conventional coralline HA to stimulate bone formation and inhibit bone resorption [73]. Nano-hydroxyapatite/coralline grafts coated with vascular endothelial growth factor (VEGF) have also been investigated and recently tested in an alveolar defect animal model leading to significantly improved neovascularization and mineralization [74]. 

Apart from corals, other traditionally investigated marine bioceramic sources are nacre, seashells (foraminifera, bivalve mollusks as oyster and mussels), sponge skeletons, diatom frustules, sea urchin spines, cuttlefish bone [61], other fish bones such as tuna [75], and shark teeth [76,77]. When obtained as powders, these ceramics showed rod-like shape particles with submicron average size as shown in Figure 1A and excellent “in vitro” biocompatibility independently of the fish source (Figure 1B) [75]. Several of the marine bioceramic sources such as certain seashells, fish bones, and shark teeth, are waste or by-products from the fishing and food industries guaranteeing source abundance. Their re-use as bone grafts contributes to sustainability, by taking advantage of this waste and re-valorizing it into products with a higher added value. Again, calcium carbonate is the composition that mainly predominates in their structure, as is the case for nacre, oyster and mussel shells, calcareous sponges’ skeleton, sea urchin spines, and cuttlefish bone [61,78]. Others are based on silicon compounds as certain sponge skeletons and diatom frustules [79]. In addition, finally, fish bones and shark teeth are sources of great interest given their composition based on calcium phosphates, as human bones [80].

From all the calcium carbonate sources, nacre is, by far, the most studied as bone graft. It consists of a highly crystallized acellular calcium carbonate powder as pseudohexagonal aragonite nanograins encapsulated into the intracrystalline organic matrix [81]. The organic material supposes around 1.7% or less [82], and could be removed by thermal treatment at around 550–600 °C, temperature at which aragonite is already transformed to calcite (it occurs at 300–400 °C) [83]. A recent review [84] collected the “in vivo” and “in vitro*”* studies which revealed its osteoinductive potential, together with osteoconductivity, biocompatibility, and biodegradability. Stimulation of new bone formation was observed when it was implanted at various sites for different uses such as human maxillary alveolar bone, load-bearing sites fillers, cranial defects fillers as well as intervertebral fusion [84,85] using different animal models [86]. This material would, therefore, solve the still existing demands for bone fillers suitable in applications that require load support, being also osteoinductive. According to Zhang et al. [84], nacre presents a great potential in the field of bone substitutes given its remarkable mechanical properties, with a Young’s modulus of 70 GPa (dry) or 60 GPa (wet) equivalent to steel [61]. The absence of nacre commercial products available for clinical use could be explained by the existence of regulatory aspects or purely commercial barriers (such as profitability) that hinder its entrance in the bone grafts market. Interestingly, there is currently a large amount of literature based on the synthesis of nacre-mimetic composites. One example is the infiltration of a thermally switchable Diels-Alder polymeric network into a lamellar scaffold of alumina, recently published by Du and coworkers [87]. 

Another promising source currently in development is the already mentioned fishing by-product shark tooth, which directly provides calcium phosphate. In fact, according to López-Álvarez et al. [76], the two sections of shark tooth, enameloid and dentine, constitute two direct sources of bioapatites, once subjected to a thermal processing to remove the organic material. Dentine offers a porous biphasic bioceramic with an apatitic phase of HA apatite-CaF in ≈60% and a non-apatitic one with whitlockite/β-TCP in ≈40%, together with a globular morphology and bimodal porosity (~50 μm and 0.5–1.0 μm). The shark enameloid provides mainly an apatitic phase of fluorapatite in 91%, as aligned elongated crystals of 0.5–1.0 μm in the shortest dimension, and a small contribution of around 9% of whitlockite/β-TCP. This enameloid bioceramic will contribute to improve bone graft mechanical properties since, according Enax et al. [88], it is about six times harder than dentine, providing high bulk moduli and stiffness constants. Moreover, the incorporation of Fluor ions into the apatite lattice will provide protection against acids [89], and is thought to have a potential role as a cell growth factor enhancer, acting primarily on the osteoprogenitor cells and/or undifferentiated osteoblasts. Thus, its presence could contribute to bone healing and regeneration by inducing the differentiation of osteoprogenitor and undifferentiated precursor cells to osteoblasts [90]. 

Moreover, the presence of fluorapatite provides low resorption levels. The resorption rate of the graft would be increased and modulated (as desired) by the contribution of the non-apatitic phase from the shark dentine. Moreover, the presence of trace elements with relevant roles in bone metabolism as Mg, together with Na, Sr, K, Al, and Fe dopes these grafts with osteoinductive properties [91]. “In vitro” [76] and “in vivo” evaluation [77] of these shark teeth grafts have confirmed their good results for bone regeneration, with higher early osteogenic activity (*p* < 0.01) on MC3T3-E1 pre-osteoblasts after 21 days of incubation than commercial synthetic and bovine bone grafts [77]. Moreover, the porosity provided by dentine together with the inter-granular cavities allowed the ingrowth of new bone cells (osteoconductive properties) after three weeks of implantation in a rodent model, showing higher osteointegration than commercial synthetic granules (biphasic HA/β-TCP (60%/40%)) [77]. Bone formation was observed from the critical defect surroundings but also at its central area indicating also potential osteoinductive properties (Figure 1D) that could be associated to the porous structure of the ceramic as shown in Figure 1C. Furthermore, significantly higher bone mineral density (*p* < 0.05) was quantified on these grafts when compared to the commercial synthetic graft [77]. Shark tooth is, therefore, an interesting source of bone grafts with great potential presenting osteoinductive properties and moldable resorption level, features that are not yet commercially available in any bone graft of biological origin. 

## 5. Concluding Remarks

In this review we summarized the current available biological-derived ceramic bone grafts, including their advantages, limitations, and applications that are collected and summarized in Table 4. Despite the wide variety of bone graft alternatives currently clinically available, there is no perfect material able to fulfill bone tissue requirements. While autografts are the gold standard, the limited volume available is their main drawback. On the other hand, allografts do not present this problem but present diminished osteoinductive and mechanical properties. The use of mammalian and marine xenografts allow for availability in large quantities but did not present the exact human bone tissue morphology and their osteoinductive capacities and biological responses are decreased when compared to autographs and allographs. 

The bone grafts market is expected to keep steadily increasing in the near future and the search for materials able to fulfill all the requirements for adequate bone function, promoting osteoinduction and osteoconduction, continues to be a point of interest for the companies. A gradual transition is emerging from current procedures based on autografts and allografts to commercial grafts of synthetic or biological origin (xenografts). However, the higher biological response obtained with natural calcium phosphate based grafts, when compared to those of synthetic origin, make them “a priori” a better choice for bone regeneration. The wide variety of marine derived ceramic grafts allowing for the second use of animal discards and fishing by-products provides a viable source of bone grafts while contributing to sustainability. The research recently performed on the marine field for new bone grafts sources has led to the discovery of new naturally-derived bone grafts with promising mechanical and biological features. However, further studies should be performed to ensure their adequate performance in terms of 3D bone regeneration and toxicity to allow their commercial exploitation and future clinical use.

## Figures and Tables

**Figure 1 marinedrugs-17-00471-f001:**
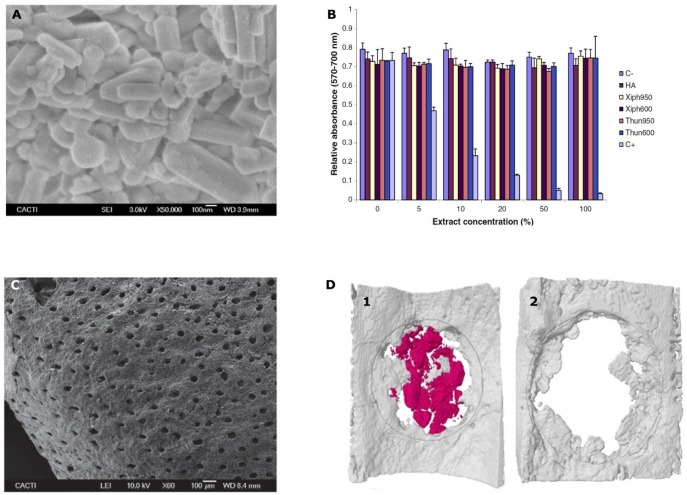
(**A**) FSEM micrograph showing the appearance of the obtained powders from fish bones at 600 °C; (**B**) cell viability of osteoblasts (MC3T3-E1) after incubation with sword fish (Xiph) or tuna (Thun) samples treated at 600 or 950 °C together with commercial hydroxyapatite (HA) and control extracts for 24 h. C+: positive control. C−: negative control; (**C**) SEM micrographs of the shark teeth bioapatites morphology at 60×; (**D**) micro-CT reconstructions of an extracted bilateral parietal rat bone defect either treated with shark teeth bioapatites (**1**) or correspondent critical defect control (**2**) after 3 weeks of implantation. Marine shark teeth bioapatites are colored in red and new bone tissue in gray. Areas of interest are delimited by gray lines. Figure 1A,B reprinted from [76] with permission from Elsevier. Figure 1C,D reprinted from [78] with permission from John Wiley and Sons.

**Table 1 marinedrugs-17-00471-t001:** Commercially available demineralized allografts (DBMs) suitable for bone defects not intrinsic to the bony structure stability ^1^.

Brand Name	Company	Compositional Details	Intended Use
Grafton^TM^ DBM	Medtronic	DBM with bone fibers, instead of particles, to create a physical network with pathways for the cells	Spine, pelvis, extremities; augment dental intraosseous, oral, and cranio-maxillofacial defects
Magnifuse^TM^ DBM Bone Graft	Medtronic	DBM fibers with surface-demineralized cortical chips in a resorbable mesh system	Spine, pelvis, extremities; Magnifuse™ II Bone Graft only for posterolateral spine and pelvis
DBX® Demineralized Bone Matrix	DePuy Synthes	DBM with sodium hyaluronate, natural derived material not of animal origin, biocompatible and biodegradable	Trauma, mandibular maxillary reconstruction, alveolar ridges, oral/maxillofacial/dental intraosseous defects, osseous defects in the cranium
Allomatrix®	Wright Medical Technology	DBM with cancellous chips containing surgical-grade calcium sulfate Osteoset^TM^	Skeletal system (i.e., extremities, spine, pelvis)
Ignite®	Wright Medical Technology	Biologic solutions that can be combined with bone marrow aspirate. Injectable	Rigid non unions. Soft and/or hard tissue repair
Alphagraft® DBM	Alphatec Spine	DBM reverse phase medium: thickens at body temperature resists irrigation to minimize the likelihood of migration from the surgical site	Designed to supplement other Alphatec Spine products
Stryker DBM	Stryker	DBM reverse phase media carrier and cancellous chips (putty plus)	Spinal procedures; oral and maxillofacial defects
io DBM	Stryker	Contains cancellous, cortical bone, and periosteum. Reverse phase medium carrier for gel, putty, and putty plus	Bone void filler or extender in posterolateral fusion procedures
AlloFuse®	AlloSource	DBM reverse phase medium: matrix easily moldable and with higher viscosity becoming thicker at warmer temperatures (human body)	General bone grafting applications in orthopedic and spinal fusion procedures
Accell Total Bone Matrix®	SeaSpine	100% DBM processed from ground cortical bone	Skeletal system as bone graft extender in spine, extremities, and pelvis, or as a bone void filler in extremities and pelvis
InterGro® DBM	Zimmer Biomet	DBM Plus with porous granules of calcium carbonate with outer layer of calcium phosphate	Extremities and pelvis, spine, craniofacial defects, craniotomies not larger than 25 cm^2^
Equivabone®	Zimmer Biomet	DBM with Etex nanocrystalline calcium phosphate technology	Bone voids or defects that are not intrinsic to the stability of the bony structure
Puros® Allograft	Zimmer Biomet	DBM in reverse phase medium. RPM Putty: with cancellous bone chips	Spinal fusion procedures and dental applications
StaGraft® Fiber	Zimmer Biomet	100% cortical fiber DBM	Orthopedic, spinal, reconstructive craniomaxillofacial, periodontal bone grafting procedures
Opteform®	Exactech	DBM with cortical cancellous bone chips (osteoconductive) and gelatin carrier	Oncology, joints, foot and ankle, hand, sports medicine, trauma, long bone fractures
Optefil®	Exactech	DBM and gelatin carrier	Oncology, joints, foot and ankle, hand, sports medicine, trauma, long bone fractures
OsteoSelect®	Xtant Medical	OsteoSelect Plus: DMB putty with demineralized cortical chips	Standalone bone graft in spinal procedures
OsteoSponge®	Xtant Medical	OsteoSponge: DBM made from 100% cancellous bone with malleable properties and shape memory	Spinal fusion devices, in arthrodesis, or in fracture sites
Progenix Putty®	Umg Uysal Medikal	DBM (70%) with type I bovine collagen (11%) as carrier and alginate (19%, dry weight)	Small or large intrabony defects through a precise 1 mm delivery syringe
Progenix Plus®	Umg Uysal Medikal	DBM and demineralized cortical chips (60%) with type I bovine collagen (5%) as carrier and alginate (35%) by dry weight. Demineralized chips provide osteoconductivity and access to growth factors	Progenix® Plus contains bone chips of approximately 2–4 mm for use in small defects
H-Genin^TM^	Berkeley Advanced Biomaterials	DBM produced from ground cortical bone	Cranio-facial surgery, spinal fusion, hand and foot surgery, fracture repair, joint reconstruction
StimuBlast®	Arthrex	DBM in reverse phase medium giving moldable properties. It thickens up at body temperature and resists irrigation	Orthopaedic applications as filler for gaps or voids that are not intrinsic to the stability of the bony structure

^1^ Compositional details and intended uses were taken from the corresponding companies’ websites.

**Table 2 marinedrugs-17-00471-t002:** Currently commercialized bone xenografts from mammal origin. The last four rows include the commercialized combinations of polymers or extracellular matrix proteins with mammalian origin xenografts ^2^.

Brand Name	Company	Compositional Details	Intended Use
Bio-Oss®	Geistlich	Bovine deproteinized bone	Periodontal, oral, and maxillofacial surgery
Orthoss®	Geistlich	Bovine derived bone substitute made from highly purified bone mineral	Filling of bone voids following trauma, reconstruction in orthopedics, and spinal surgery
Cerabone®	Botiss	Sintered bovine bone granules	Sinus lift, horizontal and vertical augmentation, ridge preservation, peri-implant defects, socket preservation, bone defect augmentation, periodontal intrabony defects, furcation defects
Endobon®	Biomet 3i LLC	Fully deproteinized bovine hydroxyapatite	Alveolar ridge augmentation, sinus elevation, filling bone defects after root resection, socket filling after tooth extraction
CopiOs®	Zimmer	Mineralized particulate cancellous bovine bone chips	Large and small bone defects
Bonefill®	Bioinnovations Inc	Natural bovine bone mineral extracted from bovine femur	Bone failure reconstructions where remodeling or bone neoformation is desired
InterOss®	Sigmagraft	Natural bovine hydroxyapatite	Bone defect filling
Apatos OsteoBiol®	Tecnoss	Heterologous cortico-cancellous bovine bone mix	Large maxillofacial bone defects, reconstruction, or corrections
GenOx Org®	Braumer	Lyophilized porous organic matrix extracted from the bovine cortical bone	Procedures of dental implant, Maxillofacial and bone surgery in general
Cerabone®	aap Implantate AG	Cancellous bovine bone	Permanent bone filling or reconstruction of aseptic bone defects
OssiGuide®	Collagen Matrix	Cancellous bovine bone	Filling bony voids or gaps of the skeletal system that are not intrinsic to the stability of the bony structure
THE Graft®	Purgo Biologics	Porcine cancellous granules	Extraction socket with intact socket, extraction socket with defective socket, minor bone, augmentation, major bone augmentation, sinus floor elevation, peri-implantitis
Gen-Os®	Tecnoss	Cortico-cancellous heterologous porcine or equine bone mix	Alveolar ridge preservation, lateral access maxillary sinus lift, dehiscence regeneration
MatrixOss®	Collagen Matrix	An organic porcine bone mineral with carbonate apatite structure	Bone filling
Sp-Block	Tecnoss	Equine cancellous bone	When a vertical gain in posterior mandible is required
BIO-GEN® BIO-GEN® MIX GEL BIO-GEN® PUTTY	BioTECK	Equine cortical bone or spongy bone	Bone regeneration surgery
Alpha-Bio’s Graft	Alpha-Bio Tec	Bovine cancellous bone + bioactive resorbable polymers	Open sinus floor augmentation, peridontal intrabony defects, peri-implant bony defects, socket preservation, vertical and horizontal bone augmentations
Bio-Oss Collagen®	Geistlich	Deproteinized bovine bone mineral small granules +10% porcine collagen	Extraction socket management, minor bone augmentation, periodontal regeneration
Gel 40 Putty mp3®	Tecnoss	Porcine or equine Cortico-cancellous heterologous bone mix + different proportions of Collagen gel	Lateral and crestal access sinus lift, deep and narrow peri-implant defects, three-wall intrabony defects, gingival recessions, post-extractive sockets, defects that present a self-contained cavity
OsteoBiol® GTO®	Tecnoss	Porcine or equine heterologous cortico-cancellous bone mix + OsteoBiol® TSV Gel	Horizontal augmentation procedures, socket preservation

^2^ Compositional details and intended uses were taken from the corresponding companies’ websites.

**Table 3 marinedrugs-17-00471-t003:** Commercially available bioceramic xenografts of marine origin.

Brand Name	Company	Compositional Details	Intended Use
BioCoral®	BioCoral Inc	Natural coral calcium carbonate wholly mineral bone graft substitute (99% calcium carbonate) [68]	Spinal surgery, tibial osteotomies, hip fractures, trephine hole replacement, fracture repair, osteoporosis; maxillocraniofacial; reconstructive and cosmetic surgery, bone defects due to loss of teeth or periodontal disease
Pro-Osteon® 200R	Zimmer Biomet	Coral calcium carbonate matrix covered by outer layer 2–10 µm thickness of calcium phosphate. Pore size 190–230 µm. Significant resorption in 6–18 months.	Indicated for bony voids or gaps that are not intrinsic to the stability of the skeletal system
Pro-Osteon® 500R	Zimmer Biomet	Coral calcium carbonate matrix covered by outer layer 2–10 µm thickness of calcium phosphate. Median pore diameter 435 µm. Significant resorption in 6–18 months.	Indicated to be gently packed into bony voids or gaps of the skeletal system (i.e., the extremities, spine, and pelvis) as for cervical fusion.
CoreBone® Coross®	Corebone/DSI, Dental Solutions Israel	Coral calcium carbonate crystals (>95%) as aragonite enriched with silicon, strontium, and other non-organic substances. Ca, Si, and Sr play important roles in bone mineralization	Maxillofacial and orthopedic indications. Interconnected porosity allows 3D generation of bone with high fusion rates, without loss of strength
Frios® Algipore®	Dentsply Sirona	Algae-derived hydroxyapatite by hydrothermal conversion of original calcium carbonate of the algae *Corallina officialis* [69]. Particle sizes 0.3–2 mm; pores of 5–10 μm [70]	Bone augmentation in the atrophic maxilla, periimplantitis lesions, alveolar ridge alteration following tooth extraction

**Table 4 marinedrugs-17-00471-t004:** Advantages, limitations, and applications of biological-derived ceramic bone grafts.

Bone Grafts	Advantages	Limitations	Clinical Application
Autografts	Osteoinductivity Osteoconductivity Biocompatibility Bone mechanical properties	Postoperative morbidity Limited volume Increase in surgical time Additional anesthetic procedure required	Gold standard in trauma and dentistry when possible
Allografts	Osteoinductivity Osteoconductivity Biocompatibility High availability Reduced surgical time	Lower osteoinductive capacity than autografts Inferior mechanical properties Costly and laborious processing Potential risk of diseases transmission	Osteoarticular reconstructive surgery Foot and ankle surgery
Mammalian xenografts	Bone tissue physiological similarities Osteogenic and bone inductive properties Excellent support for bone colonization	Low tissue remodeling Stay mainly unaltered on the host bone Batch variability	Filling of bone voids following trauma, reconstruction in orthopedics, spinal surgery, periodontal, oral, and maxillofacial surgery
Marine xenografts	Interconnected porosity Hierarchical structure Osteoconductivity Availability in large quantities	Weak mechanical properties Fast degradation Batch variability	Bone fillers in non-load bearing regions

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
