# Peer review of "Current Stage of Marine Ceramic Grafts for 3D Bone Tissue Regeneration"

_marinedrugs, 2019, doi:10.3390/md17080471_

Round 1

Reviewer 1 Report

The paper is a review of bone grafts and their sources, properties and applications. The authors discuss bone graft from both mammalian and marine origins. The review is well written, detailed, and covers the most important advances in this area. My main concern with the present manuscript is that the review is more of a survey and less of a critical review with the authors opinions. For example, the authors do represent allograft, autograft and bone xenograft from mammalian and marine origins, but it may sound simple exhibition. If the authors add one or two tables to compare their features, advantages/disadvantages, applicability, etc. this review becomes much deeper. Not limited to collection of the fact, representation of the authors through is important in review articles. Tables of the current manuscript are just fact assemblies. Furthermore, since the focus of this review is the marine ceramic graft for 3D tissue regeneration, it would be meaningful to add some figures of significant results from recent studies, highlighting the unique features, hence performances, of bone graft from marine origin. Moreover, the authors should supplement the safety and biocompatibility of these bone grafts. I recommend that the authors address this issue in the revised manuscript.

Author Response

DETAILED RESPONSES TO REVIEWERS

We would like to thank the reviewers for the careful reading of the manuscript as well as for the positive feedback given. We have made a number of modifications to the manuscript text and tables in order to address reviewers’ concerns. Detailed responses to individual reviewer comments are given below in blue. Corresponding alterations to the text have been highlighted in yellow in the manuscript.

Reviewer #1

Comment 1: The paper is a review of bone grafts and their sources, properties and applications. The authors discuss bone graft from both mammalian and marine origins. The review is well written, detailed, and covers the most important advances in this area. My main concern with the present manuscript is that the review is more of a survey and less of a critical review with the authors opinions. For example, the authors do represent allograft, autograft and bone xenograft from mammalian and marine origins, but it may sound simple exhibition. If the authors add one or two tables to compare their features, advantages/disadvantages, applicability, etc. this review becomes much deeper. Not limited to collection of the fact, representation of the authors through is important in review articles. Tables of the current manuscript are just fact assemblies.

Response 1: In agreement with the reviewer’s comment we have added an additional table (Table 4) to summarize the main advantages, limitations and applications of biological-derived bone grafts currently in clinic that has been included in the manuscript`s concluding remarks. Also, we have added a paragraph on the concluding remarks section to discuss, as a whole, all the biological-derived bone grafts.

Comment 2: Furthermore, since the focus of this review is the marine ceramic graft for 3D tissue regeneration, it would be meaningful to add some figures of significant results from recent studies, highlighting the unique features, hence performances, of bone graft from marine origin. Moreover, the authors should supplement the safety and biocompatibility of these bone grafts. I recommend that the authors address this issue in the revised manuscript.

Response 2: We have added a figure (Figure 1) representing different marine derived ceramic bone grafts morphology, biocompatibility assessments and “in vivo” bone regeneration in a critical size defect. Some of the already published data on marine grafts biocompatibility has been added to the Figure. Moreover, the companies with already on the market marine products show in their technical file their biocompatible and safe character.

Reviewer 2 Report

The authors have proposed an interesting review on compilation of current studies on marine ceramics graft for 3D bone tissue regeneration. Some minor comments are:

1)Line 47-48: Any reference?  How scaffold can facilitating oxygen when the cell in the depth center normally will face cell death due to low oxygen transport? 

2) Line 52-53: Reference and which type of patient? in dentistry or major bone fructure

Author Response

DETAILED RESPONSES TO REVIEWERS

We would like to thank the reviewers for the careful reading of the manuscript as well as for the positive feedback given. We have made a number of modifications to the manuscript text and tables in order to address reviewers’ concerns. Detailed responses to individual reviewer comments are given below in blue. Corresponding alterations to the text have been highlighted in yellow in the manuscript.

Reviewer # 2

The authors have proposed an interesting review on compilation of current studies on marine ceramics graft for 3D bone tissue regeneration. Some minor comments are:

Comment 1: Line 47-48: Any reference? How scaffold can facilitate oxygen when the cell in the depth center normally will face cell death due to low oxygen transport?

Response 1: Scaffolds are usually highly porous materials with open pores and interconnected networks that facilitate nutrient and oxygen diffusion and waste removal. These are the main advantages of scaffolds for 3D cell culture specially when compared to other alternatives for 3D cell culture as organoids or cell encapsulation. Moreover, a reference that clearly states these characteristics has been included in the manuscript.

Comment 2: Line 52-53: Reference and which type of patient? in dentistry or major bone fracture?

Response 2: Bone grafts can be used for both types of applications. However, depending on the specific type of graft the suitable clinical application varies. For more information please find a recent review on clinical applications of bone grafts (Fernandez de Grado, G. et al., Bone substitutes: a review of their characteristics, clinical use, and perspectives for large bone defects management. Journal of Tissue Engineering 2018, 9, 1-18) that has also been added as a refence on the manuscript.